# Nanocellulose-Based Polymer Composites Functionalized with New Gemini Ionic Liquids

**DOI:** 10.3390/ijms232415807

**Published:** 2022-12-13

**Authors:** Daria Zielińska, Andrzej Skrzypczak, Barbara Peplińska, Sławomir Borysiak

**Affiliations:** 1Institute of Chemical Technology and Engineering, Poznan University of Technology, Berdychowo 4, PL-60965 Poznan, Poland; 2NanoBioMedical Centre, Adam Mickiewicz University, PL-61614 Poznan, Poland

**Keywords:** nanocellulose, polysaccharide, cellulolytic enzymes, synthesis, gemini, dimeric imidazolium ionic liquids, functionalization, nanocomposites, structure, mechanical properties

## Abstract

The manuscript discusses the application of dimeric imidazolium ionic liquids with an aliphatic linker of different lengths, constituting a new class of compounds called gemini, for the modification of renewable materials. This innovative functionalization with the use of ionic liquids made it possible to obtain polymer composite nanomaterials with renewable fillers, which will reduce the consumption of petroleum-based raw materials and also be directly related to the reduction of energy intensity. Renewable filler in the form of nanocellulose modified with ionic liquids, as well as polymer composites with such filler obtained by extrusion and injection molding techniques, were subjected to detailed characterization using techniques like: X-ray diffraction (XRD), Fourier transform spectroscopy (FTIR), dispersion studies (DLS), morphological analysis (SEM), differential scanning calorimetry (DSC), hot-stage polarized light microscopy and characterization of mechanical properties. The use of innovative dimeric ionic liquids proved to be an effective method to carry out efficient functionalization of cellulose. This provided a stable space structure between polysaccharide particles, limiting aggregate formation. It was shown that chemical modification with ionic liquids has a significant effect on the nucleation activity of cellulose fillers and the formation of the supermolecular structure of the polymer matrix, which consequently allowed to obtain polymer composites with excellent strength characteristics and increased flexibility, which will allow to increase their application potential. Innovative ionic liquids have contributed to obtaining green nanomaterials with excellent functional properties, which have not been described in the literature so far.

## 1. Introduction

Polymer composites containing renewable lignocellulosic fillers are becoming increasingly popular due to their very good mechanical properties [1,2], low density or biodegradability [3]. The presented advantages make this type of material an alternative to commonly used engineering composites obtained from petroleum-based raw materials in many fields, such as the automotive, construction and packaging industries. Today, the requirement for environmentally friendly products is not only a consequence of numerous legal regulations but also increasingly driven by the need to develop technologies for the manufacture of the most environmentally friendly products, as well as in the context of the diversification of fossil resources towards renewable raw materials. Wood [4,5,6], cellulose [7,8] and its nanometric counterpart, nanocellulose [9,10,11,12], are used as lignocellulosic fillers in polymer matrix. Unfortunately, due to numerous problems with the proper distribution of cellulose particles in the polymer and the formation of agglomerates and, as a consequence, poor interfacial adhesion to hydrophobic polymers as well as the tendency of cellulose fibers to absorb water, the scientific world is looking for new solutions towards the creation of innovative plastic fillers [13].

Various methods of filler modification ranging from acetylation [14,15,16,17], esterification [18,19,20,21], etherification [22] or reaction with isocyanates [23,24,25], sulfates [26], amines [27], hemiacetals [28] and silanes [29,30,31] are used to improve adhesion between the renewable cellulose filler and the polymer matrix. However, it is worth noting that these techniques have a number of drawbacks. The solvents used to carry out the presented chemical modification reactions are toxic and expensive [32,33], and their effectiveness is limited [34].

In view of the numerous drawbacks associated with the presented modifications, which involve the use of organic solvents, the scientific world is seeking new ways to replace toxic chemicals with their more environmentally friendly and efficient replacements. A first report in the literature [35], dating to the early 21st century, proposes that an alternative to the above problem of chemical modification of cellulose could be the use of ionic liquids (ILs) which are organic salts with a melting point below 100 °C and the ability to dissolve both organic and inorganic materials [36,37]. In addition, ionic liquids are characterized by high thermal and chemical stability [22,38] and recyclability [39,40].

Heretofore, ionic liquids with various types of cations, including imidazolium, pyridine, ammonium and phosphonium in correlation with an alkaline anion [41,42], have been used to dissolve cellulose in order to produce monosaccharides from lignocellulosic biomass [43,44,45]. Swatolski et al. [35] used imidazolium 1-butyl-3-methyl cation ionic liquids with different anions and found that chloride, which acted as a hydrogen bond acceptor, was the most effective anion for dissolving polysaccharides than its non-coordinating equivalents. Moreover, in another study, the researchers investigated the effect of structure on the efficiency of cellulose dissolution through ionic liquids and proved that 1-ethyl-3-methylimidazolium diethyl phosphate was the most effective substance among other tested organic salts against cellulose dissolution reactions. In addition, they justified the very good effect of this compound by its low melting temperature, which has a direct impact on the efficiency of the reaction [46]. On the other hand, FitzPatrick et al. [47] found that their proposed ionic liquid (1-ethyl-3-methylimidazolium acetate) dissolves cellulose with high efficiency, although the viscosity of the used organic salt has a key effect on the overall process. In addition, Sayyed et al. [48] claim that the anions included in ionic liquids, which are strong hydrogen acceptors, are most effective in dissolving lignocellulosic substances. Another of the interesting applications of ionic liquids is their use as biocides or impregnating agents for lignocellulosic materials. One example of the use of liquid organic salts is the work of Sarvaramini et al. [49], where the authors impregnated lignocellulosic materials to obtain a highly hydrophobic renewable substance. On the other hand, Pernak et al. [50] using 3-alkoxymethyl1-methylim-idazolium tetrafluoroborates and hexafluoro-phosphates obtained a polysaccharide with very efficient biocidal properties.

It is worth noting that ionic liquids can also be applied as solvents in functionalization reactions of cellulosic materials to improve their physicochemical properties. An example of the application of the reactants is the use of diallyl carbonate with 1-butyl-3-methylimidazolium chloride as a solvent in the alkoxycarbonylation of cellulose, thereby obtaining a material with improved thermal properties. In addition, the researchers state that the reactants used in the functionalization of the polysaccharides with full efficiency can be reused by recycling [51]. Ma et al. [52] used 1,4-dibutyl-3-methyl-1,2,3-triazolium acetate to carry out chemical modifications of cellulose. Heinze et al. [53] also successfully used 1-N-butyl-3-methylimidazolium chloride in the synthesis of cellulose acetate with a high degree of polymerization.

The use of ionic liquids as non-direct modifiers of lignocellulosic materials without the use of solvents seems extremely interesting. One of the works that focuses on the use of organic salts without the use of additive solvents is the study of Borysiak et al. [4], whose lignocellulosic filler (*Pinus sylvestris* L.) modified with didecyldimethylammonium bis(trifluoromethylsulfonyl)imide in combination with a polypropylene matrix showed increased nucleation activity, and the polymer composite had better strength parameters than the pure polymer matrix. Moreover, Odalanowska et al. [54] using ammonium and imidazolium ionic liquids to functionalize wood proved that a properly carried out reaction, taking into account the structure of ionic liquids, is necessary to obtain plastic fillers characterized by better nucleation and strength parameters. In addition, the researchers noted that the structure of the used organic salts has a direct effect on the supermolecular structure of polypropylene composites. This is extremely significant in the processing of the presented materials. In turn, other researchers used N-hexylpyridinium acetate and N-hexylpyridinium trifluoroacetate as modifiers of microcrystalline cellulose (MCC), thus obtaining effective dispersing solvents for polysaccharides [55], while the use of 1-n-butyl-3-methylimidazolium chloride at different concentrations to functionalize MCC resulted in improved thermal stability of the filler without degrading its structure [56]. Subsequently, Croitoru et al. [57] functionalized wood methyltrioctylammonium bis(trifluoromethylsulfonyl)imide and trihexyltetradecylphosphonium bis(2,4,4-trimethylpentyl) phosphinate to produce an interesting thermoplastic matrix filler. The obtained polyethylene composites were characterized by better mechanical properties and increased stability of the composite material to water.

The present study attempts to synthesize dimeric imidazolium ionic liquids with an aliphatic linker of different lengths, constituting a new class of compounds called “gemini”, which have been used as innovative modifiers of nanocellulose obtained by controlled bioconversion of cellulose with a cellulolytic enzyme derived from a microorganism of the genus Trichoderma. Therefore, designed organic salts will effectively functionalize cellulose nanofiller by forming a stable space structure between polysaccharide particles, ensuring at the same time a reduction or elimination of the formation of non-cohesive aggregates. To our knowledge, the design and application of this type of ionic liquid, gemini, for cellulose functionalization have not been reported before. This will be the first literature report in this research area. According to our assumptions, this novel method of cellulose functionalization will directly affect the formation of the supermolecular structure of the polymer matrix, as well as the mechanical properties of the obtained polymer composites, which will allow for increased application potential.

## 2. Results and Discussion

### 2.1. Characterization of Nanofillers

#### 2.1.1. Supermolecular Structure of Obtained Nanocellulosic Fillers

XRD studies were used to characterize the supermolecular structure of cellulose nanofillers obtained by enzyme hydrolysis reactions (C) and fillers functionalized with imidazolium ionic liquids (C-IL1; C-IL2). Figure 1 illustrates the diffractograms of the analyzed materials.

Diffractometric analyses confirmed the existence of a polymorphic variety of cellulose I. The existence of diffraction maxima at 2θ angles of 15°, 17°, 22.5° and 35° originating from lattice planes (110), (110), (200) and (004) is characteristic for polymorph I [58].

Structural studies showed that carrying out chemical modification with imidazolium ionic liquids did not cause changes in both crystalline and amorphous structures. In addition, all samples exhibit comparable intensities of diffraction maxima.

In the literature, ionic liquids have most often been used to hydrolyze cellulose to obtain glucose as a degradation product of the polysaccharide [59,60,61] or to obtain a cellulose filler with nanometric particle size [62,63,64]. The researchers proved that this modification caused pronounced changes in the degree of crystallinity of the analyzed materials by “attacking” the ionic liquids on the amorphous regions of cellulose. It is noteworthy that in the present study similar action of ionic liquids was not found as described by other researchers, it means that no visible changes in the supermolecular structure of the functionalized cellulose materials were obtained, and consequently no differences in the degree of crystallinity of the analyzed samples were found. The explanation for this effect of imidazolium ionic liquids may be that there was an effective reaction between the carboxyl groups of gemini ionic liquids and the hydroxyl groups (-OH) of cellulose, without the interference of ILs with the glycosidic bonds of the polysaccharide. In order to confirm the validity of the presented thesis, Fourier transform infrared spectroscopy analyses were carried out in a further stage of the study (Section 2.1.2).

#### 2.1.2. FTIR of Fillers

Fourier transform infrared spectroscopy (FTIR) was used to determine the effectiveness of chemical modification of the cellulosic material with imidaziolium ionic liquids. Figure 2 shows FTIR spectra before and after chemical modification of the polysaccharide.

The FTIR spectra obtained for cellulose nanofillers allowed confirmation of the efficiency of chemical hydrolysis using imidazolium ionic liquids and proved changes in the chemical structure of the tested nanocelluloses. The studies were conducted in the range of 4000 to 500 cm^−1^. Significant changes in the chemical structure were observed for nanocellulose modified with ionic liquids, both with a longer alkyl linker (C-IL2) and with a shorter alkyl linker (C-IL1). A decrease in intensity and narrowing of the band, as well as a shift toward lower values of the wave number, were found for the vibration of the hydroxyl group (-OH) at around 3350 cm^−1^. Bands at a wave number of 3350 cm^−1^ can also be attributed to bending vibrations for internal and intermolecular hydrogen bonds. Differences in the intensity of the bands characteristic of both hydroxyl group vibrations and bending vibrations can be related to changes in the number and strength of the hydrogen bonds [65]. In addition, these changes in the intensity of the wave band at 3350 cm^−1^ can be attributed to the reaction of the hydroxyl groups present in the cellulose material with the carboxyl groups existing in the used ionic liquids. Small changes in the intensity of the bands at 2900 cm^−1^, corresponding to C-H stretching vibrations, were also found for celluloses hydrolyzed with ionic liquids [66]. Pronounced changes were observed at a wave number of 1740 cm^−1^ a band derived from stretching vibrations for the carbonyl group (C=O), which is undeniable evidence of the efficiency of the modification reaction carried out between the carboxyl groups present in the ionic liquids and the hydroxyl groups in the cellulosic material. It is worth noting that a slightly higher intensity of this band was characterized for the C-IL1 system. A decrease in the intensity of CH deformation bands at 1370 cm^−1^, CH2 bands at C-6 for 1316 cm^−1^ and COH oscillations at C-6 for 1199 cm^−1^ for nanocelluloses treated with ionic liquids was also observed. A decrease in the relative absorbance of the bands (radiation absorption) is also noticeable for the deformation vibrations of the CH2 groups at C-6, as well as for OCH at 1429 cm^−1^. The deformations, dislocations and twists of the vibrations characteristic of anhydroglucopyranose in the band from 1800 cm^−1^ to 600 cm^−1^, which relate to β-glucosidic bonds, testify to the occurrence of the first polymorphic variety—cellulose I [66,67].

The presented changes in the chemical structure of cellulose nanofillers confirm the effectiveness of the performed chemical modification with imidazolium ionic liquids, as evidenced by the acquisition of additional bands derived from carbonyl bonds (C=O) and a reduction in the intensity of bands derived from hydroxyl groups.

#### 2.1.3. Dispersive Properties of Obtained Fillers

The particle size of the obtained cellulose nanofillers was analyzed based on the implementation of the dynamic light scattering technique. Figure 3 summarizes the results of the DLS analysis for the tested nanomaterials.

Unmodified nanocellulose obtained by enzymatic hydrolysis with cellulase from the microscopic fungus *Trichoderma reesei* was characterized by the presence of two types of particles in the range from 59 to 106 nm for the nanometric fraction (in the amount of 71%) and in the range from 1.72 to 3.06 μm for micrometric particles (in the amount of 29%). In the nanometric fraction, the largest percentage of nanoparticles was characterized by grains with a diameter of 79 nm (27.0%), while on the micrometric scale the largest percentage of these segments could be observed at 2.30 μm (11.9%).

The results of the dispersion analysis are different with respect to the analyzed nano-celluloses treated with dimeric imidazolium ionic liquids, in which only one type of particle is present—the nanometer fractions in the range from 51 to 106 nm. It was found that the nanofiller, which was obtained using an ionic liquid characterized by a shorter alkyl linker (C-IL1) was characterized by a dominant contribution of nanoparticles with a diameter of 79 nm at 37.5%, while for the filler modified with an ionic liquid with a longer alkyl linker—C-IL2, particles with a diameter of 69 nm provided a maximum volume contribution of 35.1%.

Dispersion studies confirmed that performing chemical modification with ionic liquids has a significant effect on the dispersion properties of the resulting cellulose nanoparticles. It was found that the application of the chemical functionalization method resulted in a polysaccharide characterized only by nanometric particle size, while its unmodified counterpart was characterized by the presence of both nano- and micrometric particles. The explanation for obtaining a material with two types of particles in an unmodified polysaccharide, sample (C), may be the presence of a large number of hydroxyl groups (-OH), which are capable of forming agglomerates due to intermolecular interactions, such as through hydrogen bonds [68,69]. As stated earlier, cellulosic materials treated with dimeric imidazolium ionic liquids were characterized by the occurrence of only the nanometer fraction, which may be due to the formation of permanent covalent bonds between cellulose molecules as a result of the functionalization reaction carried out with ionic liquids. Such a reaction mechanism simultaneously minimizes or eliminates the unfavorable intermolecular bonds responsible for the aggregation of polysaccharide nanoparticles and the formation of a stable space structure, similar to cross-linked structures. The likely mechanism of action of the used ionic liquids is illustrated in Figure 4.

#### 2.1.4. Morphological Characteristics of Analyzed Nanomaterials

Scanning electron microscopy (SEM) was used to define the surface morphology of the obtained nanofillers as well as to determine the shape of the polysaccharide particles. Figure 5 illustrates the microscopic images of the analyzed cellulosic materials.

Microscopic analyses ideally indicate that the tested cellulosic material is characterized by the spherical shape of the nanoparticles. It is noteworthy that the obtained nanomaterials are typified by the presence of particles whose size is less than 100 nm, which was also confirmed in earlier DLS studies (Section 2.1.3). It was found that the particle size for nanocellulose obtained by enzymatic hydrolysis reaction (C—(a)) is about 40–56 nm, while the sizes for the materials additionally modified with imidazolium ionic liquids are comparable and rank between 25 and 53 nm.

It is well known that obtaining ideal spherical nanofiller particles is an extremely difficult challenge. Scientists emphasize that in order to produce spherical CNC particles, advanced chemical techniques involving the use of concentrated acids are required, which are preceded by supporting processes that include, for example, a method using ultrasound [70]. It is also worth signaling that the application of ionic liquids, although lacking functional groups capable of reacting with the hydroxyl groups of cellulose, contributes to the formation of fibrillar cellulose nanoparticles [71]. In our work, functionalization of cellulose with gemini ionic liquids did not result in changes in the shape and size of the nanoparticles. This proves that the ionic liquids used do not affect the course of the hydrolysis reaction of polysaccharide chains and, consequently, change the nanometric form of cellulose, as confirmed by the lack of changes in the oscillatory spectrum derived from the glycosidic bond. The applied technique of functionalizing cellulose with dimeric imidazolium ionic liquids is selective and concerns only the course of the esterification reaction between the hydroxyl groups of cellulose and the carbonyl groups present in the ionic liquids, as confirmed by FTIR studies (Figure 2). In summary, the results presented are compatible with the previously discussed structural studies, FTIR spectroscopy and particle size analyses.

### 2.2. Characterization of Nanocomposites

#### 2.2.1. Analysis of Phase Transition

Differential scanning calorimetry (DSC) studies were used to determine the phase transitions occurring in the tested nanocomposite materials. Figure 6 shows thermograms derived for the crystallization process of pure polypropylene matrix in the presence of cellulose nanofillers. In addition, Table 1 summarizes the determined characteristic parameters for the analyzed polypropylene nanocomposites—crystallization temperature (T_c_), melting temperature (T_m_) and half-time of crystallization (t_0.5_) calculated from the phase conversion rate (α).

The crystallization temperature for the pure polypropylene matrix is located at 114.6 °C. It was observed that the addition of cellulose nanofillers to the polymer matrix significantly changes the position of the exothermic peaks, towards higher crystallization temperatures. The highest increase in T_c_ values by 12 °C was observed for the nanocomposite material with cellulose filler obtained by an enzymatic reaction with cellulase from a microorganism of the genus Trichoderma (PP + C). Slightly lower values of crystallization temperature (121.3 °C) were obtained for the nanocomposite system with cellulose chemically modified using an ionic liquid with a longer alkyl linker (PP + C-IL2), while the lowest increase in crystallization temperature of 118.3 °C was observed for the nanocomposite with a filler obtained by reaction with the second of the used chemical modifiers (PP + C-IL1).

Similar conclusions regarding calorimetric studies were reached by researchers [4], who also used ionic liquids as modifiers of the lignocellulosic filler, where after adding the obtained filler to the polypropylene matrix, they achieved systems characterized by an increase in the crystallization temperature relative to the pure polymer matrix.

Analyzing the results of endothermic transitions, it was found that the type of nanocellulose used as a plastic filler did not affect the obtained values of melting temperatures. Moreover, the T_m_ were comparable and ranked at around 165 °C.

Figure 7 shows the curves of the degree of phase conversion (α), from which the half-times of the crystallization process (t_0.5_) summarized in Table 1 were determined sequentially.

The unfilled polypropylene (PP) matrix had the highest half time of 2.6 min, which is consistent with other literature reports [72]. It was found that for all the analyzed nano-composite systems, higher values of the phase conversion rate and, consequently, lower parameters of half-times were obtained than for the pure PP matrix. The shortest t_0.5_ (1.4 min) was observed for composites with cellulose filler after hydrolysis with cellulolytic enzyme from the microorganism *Trichoderma reesei* (PP + C). Similar values of this parameter were obtained for nanocomposites whose filler was modified with imidazolium ionic liquids. The composite system (PP + C-IL1) with the filler given a chemical reaction with an ionic liquid with a shorter alkyl linker had a half-time of crystallization equal to 1.6 min, while an increase in this parameter to 1.8 min was recorded for the competing material (PP + C-IL2).

Odalanowska et al. [54] attempted to modify pine wood using imidazolium ionic liquids and proved that the length of the alkyl substituent of the used modifiers has a significant effect on the crystallization processes of the polypropylene matrix against the lignocellulosic fillers and on the obtained half-time of crystallization of the tested materials.

Analyzing the presented results, it can be concluded that the chemical modification of the cellulose filler with dimeric imidazolium ionic liquids affects the nucleation activity of the produced nanocomposite systems. To confirm the above considerations, the following part of the study was undertaken to determine the effect of the type of nanofiller on the nucleation ability of the polypropylene matrix by defining the induction times, the density of spherulitic structures and the growth rate of the transcrystalline structure in the analyzed samples.

#### 2.2.2. Analysis of Crystallization Process

Nucleation of an isotactic polypropylene matrix in the presence of natural cellulose nanofillers was carried out by polarized light microscopy (PLM). Figure 8 shows the nucleation process at 136 °C of polymer nanocomposites in the presence of nanocellulose fillers.

Microscopic images illustrate the formation of spherulitic structures in the presence of nanocellulose fillers. It is worth noting that all nanocomposite materials are characterized by the formation of transcrystalline structures at the polymer-filler interface, but with different efficiencies. The first of the analyzed systems with unmodified cellulose filler (PP + C) is characterized by a very high nucleation capacity. In addition, this material had a significantly higher nucleation density than the other analyzed materials. Microscopic analyses showed that chemical modification of cellulose with an ionic liquid with a shorter alkyl linker (PP + C-IL1), as well as with a longer linker (PP + C-IL2), resulted in a marked reduction in the nucleation ability of the filler in the polypropylene matrix.

As is known, the formation of transcrystalline structures is associated with the occurrence of interactions between the filler and the polymer matrix. Scientists note that because of the chemical modifications carried out, very often the filler is characterized by a reduced ability to nucleate in the polymer matrix [73,74,75], which they justify by removing, for example, impurities during filler processing. An interesting conclusion was reached by Borysiak et al. [4], who treated wood chemically with ammonium ionic liquids and obtained a plastic filler characterized by a significantly lower density of the transcrystalline layer in the tested composite systems than polymers with unmodified wood. On the other hand, Odalanowska et al. [54] showed higher nucleation activity of fillers functionalized with imidazolium ionic liquids than their native counterparts.

Microscopic analysis was also used to determine the kinetic parameters of the crystallization process of the tested samples. Table 1 summarizes the induction times and the calculated growth rate of the transcrystalline structure (TCL). The pure polypropylene (PP) matrix was characterized by the longest induction time of 5 min. In addition, this system was characterized by a significantly prolonged growth rate of the transcrystalline structure. The shortest induction time (1 min) and fastest growth of spherulites (2.7 μm/min) were observed for the nanocomposite with cellulose filler obtained by an enzymatic reaction with cellulase of the microorganism Trichoderma (PP + C). In contrast, for systems with cellulose nanofiller modified with imidazolium ionic liquids, the growth rate of the transcrystalline structure was significantly reduced and was 1.7 μm/min for PP + C-IL1 and 1.3 μm/min for PP + C-IL2, respectively. In addition, using chemical modification with an ionic liquid with a longer substituent, a plastic filler characterized by an extended induction time (PP + C-IL2) to the level of 4 min was obtained. It is worth mentioning that the formation of TCL structures is influenced by various factors, ranging from the temperature at which the crystallization process is carried out to the cooling rate or the type of filler that is used [75,76,77,78].

The microscopic results are consistent with the previously discussed calorimetric tests, in which it was shown that nanocomposites with cellulose filler modified with imidazolium ionic liquid with a shorter alkyl linker (PP + C-IL1) and with a longer linker (PP + C-IL2) showed lower nucleation activity than the system with enzymatically hydrolyzed nanofiller (PP + C), which is confirmed by the determined kinetic parameters of the nucleation and crystallization processes of the isotactic polypropylene matrix relative to the used renewable fillers. As is well known, matching the crystal structure between cellulose and polypropylene can initiate epitaxial growth of polypropylene, which consequently determines the heterogeneous nucleation and the growth of the transcrystalline structure in the polypropylene matrix. Carrying out chemical modification with ionic liquids probably disrupted the crystallographic fit between the filler and the polypropylene matrix, resulting in reduced nucleation activity. In addition, chemical functionalization of nanocellulose caused space binding of nanocellulose particles, which is a consequence of the location of covalent bonds between polysaccharides. The space-modified cellulose material can be expected to be a rigid system, the presence of which may impede the mobility of polypropylene chains between nanocellulose particles and, as a result, reduce nucleation activity. The mechanism explaining the action of gemini ionic liquids against polysaccharide modification is shown in Figure 4 in Section 2.1.3.

#### 2.2.3. Supermolecular Structure of Nanocomposite Materials

Structural changes in the pure polypropylene matrix and the tested nanocomposite materials were analyzed using X-ray diffraction (XRD). Figure 9 illustrates X-ray diffractograms for the polymer matrix and polypropylene nanocomposites.

The summarized diffractometric curves illustrate the occurrence of two polymorphic varieties in the polypropylene material—the unilinear α form (2θ = 14°, 17°, 18.5°, 21° and 22°) as well as the pseudohexagonal β variety (2θ = 16.2°). In Figure 9, at the diffraction maximum originating from the (300) plane, are the noticeable significant differences in the content of the β-PP form. The content values of the β-PP form are summarized in Table 2.

It was found that for the pure polypropylene matrix, the content of the β-form was at the level of 9%, while the introduction of cellulose nanofillers increased the value of this pseudohexagonal variety in all analyzed nanocomposites. In the case of using nanocellulose obtained by enzymatic hydrolysis (PP + C), the nanocomposite material was characterized by a β-PP equal to 27%. In contrast, for the PP + C-IL1 and PP + C-IL2 systems, the content of the β-PP form ranged from 44 to 49%, respectively.

According to the literature, the formation of β-PP variety can be caused, for example, by the application of shear forces during the processing of isotactic polypropylene matrix [79]. The introduction of fillers into the polymer increases the shear stress at the polymer-filler interface, which affects the formation of significant amounts of β-PP variety. On the other hand, one factor for this large variation in structural studies in polymer composites may be the dispersion-morphological parameters of the filler. Dynamic light scattering studies are discussed in Section 2.1.3. indicate the presence of only nanometer-sized particles, not exceeding the size of 106 nm for fillers treated with gemini ionic liquids. Although in the case of unmodified nanocellulose, two types of filler particles were found—nano and micrometric. It can be concluded that the occurrence of a greater number of nanometric cellulose particles in the tested nanocomposites (PP + C-IL1 and PP + C-IL2) certainly contributes to the generation of significantly higher shear forces in the processing, as well as increased interfacial interactions between the filler and the polymer, resulting in a higher content of the β-PP variety in the analyzed samples. The rationale for this may be the carried out effective functionalization of the nanofiller through the formation of a stable space structure between the polysaccharide particles (Figure 4), which, when flowing in the molten polymer matrix during processing processes, will affect the intensification of interactions at the polymer-filler interface and, consequently, lead to the generation of the β-PP polymorphic variety.

#### 2.2.4. Mechanical Properties

Mechanical tests of the polypropylene matrix and the obtained nanocomposite systems were carried out to determine the effect of the type of filler additive on the strength parameters of the analyzed materials. The results obtained for the mechanical properties of the tested samples are presented in Table 2 in Section 2.2.3.

The obtained findings from the mechanical tests proved that the introduction of nanometric cellulose fillers into the isotactic polypropylene matrix caused pronounced changes in the strength properties of the analyzed materials. It was noted that the addition of cellulose obtained by enzyme reactions, as well as functionalization with ionic liquids, contributed to an increase in the value of the tensile strength of the nanocomposites. The lowest value of this parameter, at about 29 MPa, was reached for the pure polypropylene (PP) matrix, while an increase in this value was observed for each of the analyzed nanocomposites. For the polymer material with the addition of unmodified nanocellulose (PP + C), the value of this parameter increased by about 10% and was at a level of about 33 MPa. Noteworthy is the increase of this parameter by about 8–10 MPa for the other analyzed nanocomposites with functionalized nanofiller (PP + C-IL1 and PP + C-IL2) relative to the pure polypropylene matrix.

A similar relationship was observed for another of the studied parameters, Young’s modulus. It was proven that the introduction of innovative fillers also resulted in an increase in the value of this parameter for each of the analyzed nanocomposites. The highest Young’s modulus of about 1.70 GPa was characterized by the composite material, the filler of which was obtained by functionalization using ionic liquid with a longer alkyl linker (PP + C-IL2). The effect of treating lignocellulosic fillers with ionic liquids was also analyzed by Odalanowska et al. [54], who obtained polymer composites characterized by improved strength parameters such as tensile strength and Young’s modulus.

Similar results were obtained for each of the analyzed samples for the parameter of elongation at break. The value of this parameter for the pure polypropylene (PP) matrix was at a level of about 300%, while the addition of nanocellulose obtained by an enzymatic reaction (PP + C) did not cause significant changes in the analyzed parameter. For systems with nanofiller functionalized with dimeric imidazolium ionic liquids, it was found that the presence of a polysaccharide obtained by chemical modification with an ionic liquid with a shorter as well as a longer alkyl linker was characterized by an increase in this parameter of about 90% for PP + C-IL1 and about 197% for PP + C-IL2 with respect to the polymer system with the addition of cellulose after enzymatic treatment (PP + C).

Tensile tests proved that all composite materials had a lower impact strength of about 11–19 kJ/m^2^ compared to the unfilled polymer matrix. The lowest value of this parameter, 29 kJ/m^2^, was reached for the system with unmodified nanocellulose (PP + C); however, for composites containing nanometric cellulose subjected to chemical modification with gemini ionic liquids—PP + C-IL1 and PP + C-IL2, higher impact strength values were recorded at the levels of 34 kJ/m^2^ and 38 kJ/m^2^, respectively, compared to the material with unmodified filler.

It is worth noting that chemical modification with gemini ionic liquids ensured the introduction of rather large imidazole rings into the polysaccharide particles, whose use as fillers in the polypropylene matrix resulted in a certain “plasticizing effect”. However, it should be emphasized that the explanation for the high flexibility of such composite materials should also be interpreted by taking into account other factors, such as proper dispersion of the cellulose nanofiller in polymer matrix, minimizing the formation of defects, as well as obtaining a very large amount of the polymorphic β-PP variety, which is characterized by high flexibility compared to the α-PP variety.

The mechanical tests closely correlate with the structural analyses of the composite materials that were carried out earlier. It was shown that the addition of polysaccharide nanofillers subjected to chemical modification with ionic liquids caused a marked increase in the content of the polymorphic β-PP variety, which is reflected in the dispersion-morphological parameters of the analyzed nanofillers. In addition, the increase in the content of the pseudohexagonal variety in the polymer composites is responsible for growing the value of elongation at break, as well as improving impact resistance [80,81].

## 3. Materials and Methods

Micrometric cellulose material with trade name Sigmacell Type 20 with an average particle size of 20 μm, cellulolytic enzymes from the microscopic fungus *Trichoderma reesei* with activity ≥ 700 U/g (ATCC 26921), and all reagents used for the synthesis of designed ionic liquids were purchased from Sigma-Aldrich (Poznań, Poland). Isotactic polypropylene PP (MFR230 °C/2.16 kg—3 g/10 min) with the trade name TATREN HT 3 06 from Slovnaft (Bratislava, Slovakia) was used as a polymer matrix.

### 3.1. Controlled Enzymatic Bioconversion of Cellulose

The process of controlled enzymatic hydrolysis of cellulose was carried out in two steps. The first was a 30-min preincubation of cellulose material in citrate buffer at 50 °C. Subsequently, using cellulolytic enzymes obtained from the microscopic fungus *Trichoderma reesei*, the material after pre-incubation was subjected to the actual enzymatic reaction for 24 h under the same temperature conditions. The obtained nanometric cellulosic material was washed ten times with deionized water and dried in a laboratory dryer (BINDER GmbH, Tuttlingen, Germany) at 100 °C for 12 h.

### 3.2. Synthesis of Ionic Liquids

The first step of the synthesis, carboxymethyl-imidazolium chloride, was prepared according to the methodology described by Yahya [82]. In the next step, based on the synthesis method developed by Kaczmarek [83], 3,3′-[1,4-butane]-bis(1-carboxymethylimidazolium) dichloride was obtained by reacting carboxymethyl-imidazolium chloride (0.2 mol) with the corresponding dichloroalkanes (0.1 mol). The reaction was carried out for 24 h at 60 °C, and acetonitrile was used as a solvent. After the reaction, the solvent was evaporated. Then, hexane was added to the synthesized salt to remove impurities. The precipitated reaction product was isolated using a filtration process. Then, 3,3′-[1,4-butane]-bis(1-carboxymethylimidazolium) dichloride was dried under reduced pressure (70 °C) for a period of 24 h. Subsequently, a saturated aqueous solution of 3,3′-[1,4-butane]-bis(1-carboxymethylimidazolium) dichloride was added to a saturated aqueous solution of an inorganic salt (lithium bis(trifluoromethanesulfonyl)imide (Li[NTF_2_]) at a molar ratio of inorganic salt to organic salt of 2:1. After the metathesis reaction, the solution was decanted, and the final product was washed with distilled water and dried under vacuum Memmert UF450 (Memmert GmbH + Co. KG, Schwabach, Germany) at 50 °C for 48 h [84].

The second imidazolium ionic liquid with a 3,3′-[1,8-octane]-bis(1-carboxymethylimidazolium) cation was prepared analogously to the synthesis of the ionic liquid presented above.

The synthesized dimeric imidazolium ionic liquids are illustrated in Table 3, along with the designations used and their chemical formulas.

### 3.3. Functionalization of Cellulose Using Ionic Liquids (Chemical Modification)

The nanocellulose obtained by the enzymatic process (Section 3.1.) was functionalized using previously designed and synthesized dimeric imidazolium ionic liquids. Figure 10 illustrates the scheme of chemical modification with gemini ionic liquids.

The first step was to dissolve the ionic liquid in a binary flask at 125 °C. Subsequently, nanocellulose was added to the solution at a mass ratio of 1:8. The reaction of chemical modification of nanometric cellulose was carried out for 2 h at the set temperature conditions (125 °C) with simultaneous stirring of the solution at 150 rpm/min. After the reaction, the resulting suspension was washed with acetonitrile to remove unreacted ionic liquid. The modified nanocellulose material was dried in a laboratory dryer (BINDER GmbH, Tuttlingen, Germany) at 110 °C for 30 min. The process of chemical modification of the polysaccharide for the second liquid was carried out equally. In addition, solvent (acetonitrile) recovery was carried out in a pressure evaporator (BÜCHI AG, Flawil, Switzerland). The inventory of the obtained cellulosic nanomaterials is shown in Table 4.

### 3.4. Characterization of Synthesized Nanofillers

To characterize the supermolecular structure of the modified nanofillers, an X-ray diffraction (XRD) method was applied using a SmartLab SE diffractometer (Rigaku, Tokyo, Japan). An X-ray equipped with a copper anode with a wavelength (Cu Kα) of 1.5418 Å (40 kV; 30 mA) was used as the X-ray source. Measurements were carried out in the 2θ angle range from 5° to 40° with a counting step of 0.04°(2θ).

Structural changes of the nanometric nanofillers were determined based on the Fourier transform infrared (FTIR) spectroscopy technique using a Nicolet iS5 FTIR spectrometer (Thermo Fisher Scientific, Madison, Waltham, MA, USA). In view of the used technique, a sample was prepared in lozenge form by mixing 1 mg of cellulose with 200 mg of potassium bromide (KBr, Sigma Aldrich, Darmstadt, Germany). The measurement (16 scans) was carried out in the range of 4000 to 500 cm^−1^ with a resolution of 4 cm^−1^.

The surface morphology of the nanocelluloses was characterized using scanning electron microscopy (SEM) with a JEM-7001F TTLS microscope (JEOL Ltd., Tokyo, Japan).

Particle size and dispersion characteristics of the cellulose nanofiller were carried out based on the application of the dynamic light scattering (DLS) technique based on non-invasive backscattering (NIBS) using a Zetasizer Nano ZS-90 device (Malvern Instruments Ltd., Malvern, UK). A 1 mg sample was prepared and dispersed in propanol (25 cm^3^). The material was then homogenized using ultrasound for 20 min.

### 3.5. Production of Polypropylene Nanocomposites

The production of polypropylene nanocomposites with functionalized nanocellulose fillers was carried out in two processing steps—extrusion (I) and injection molding (II). In the first stage (I), nanocellulose and isotactic polypropylene were mixed using a twin-screw extruder Zamak 16/40 EHD (Zamak Mercator, Skawina, Poland) with a plasticization temperature ranging from 160 °C to 200 °C, the temperature of which depended on the zones of the device, and a screw speed of 150 rpm. In the second stage (II), the granules were injection molded using an ENGEL 80/25 HLS machine (ENGEL Austria GmbH, Schwertberg, Austria) with a mold temperature of 30 °C and an injection speed of 90 mm/s. The final products were moldings with 1 wt. % nanofiller.

### 3.6. Characterization of Nanocomposites with Functionalized Nanofillers

Phase transitions in polypropylene nanocomposites were defined using the differential scanning calorimetry (DSC) technique with a DSC 200 calorimeter (Netzsch, Selb, Germany). Duplicate measurements of each sample were carried out in an inert gas environment—nitrogen. The analysis was carried out in two temperature cycles: heating at a rate of 10 °C/min in the temperature range from 40 °C to 200 °C and cooling at a rate of 5 °C/min from 200 °C to 40 °C. Analyzing the obtained thermograms, the necessary kinetic parameters, such as the crystallization temperature (T_c_), melting temperature (T_m_), enthalpy of the melting process (ΔH_m_), as well as the half-time of crystallization (t_0.5_), were determined.

The isothermal process of the polypropylene matrix in the presence of the modified cellulose nanofillers was observed using the polarized light microscopy (PLM) technique with a Labophot-2 polarized optical microscope (Nikon, Tokyo, Japan) equipped with a Linkam TP93 heating attachment (Linkam, Red Hill, UK). The test was conducted in two stages—heating at a rate of 40 °C/min to 200 °C and cooling at a rate of 20 °C/min to 136 °C. Microscopic images were taken with a Panasonic CCD vision camera (Panasonic, Japan) every 30 s after the sample reached 136 °C. Microscopic analyses allowed defining the following kinetic parameters—induction time and growth rate of transcrystalline structure on the surface of nanofillers.

The supermolecular structure of the composite materials was defined using X-ray diffraction (XRD). The same measurement conditions were used as for the analysis performed against the obtained nanofillers (Section 3.4) in the 2θ angle range from 10° to 30°. After the preparation of the diffractograms (separation of the curves and determination of the amorphous part and baseline), the content of polymorphic varieties of the polymer—α and β forms—was determined using the Turner–Jones Formula (1) [85], where k is the proportion of the β-PP variety in the polymer matrix.
(1)$$k=\frac{I_{\beta 1}}{I_{\beta 1}+\left(I_{\alpha 1}+I_{\alpha 2}+I_{\alpha 3}\right)}$$

In the above formula, the designations I_α1_+I_α2_+I_α3_ correspond to the peak intensities from the polymorphic form, α-PP, while I_β1_ is the intensity of the peak coming from the β-PP variety.

Mechanical testing (tensile strength, Young’s modulus and elongation at break) of the obtained polypropylene nanocomposites in the form of moldings was carried out according to PN-EN ISO 527-3:2019-01 using a Zwick Z020 testing machine (Zwick/Roell, Ulm, Germany). The impact strength test was provided as stated in PN-EN ISO 179-2:2020-12 using Charpy’s hammer CEAST 9050 (Instron, Norwood, MA, USA) in the same sample form. The composite specimens in the number of ten repetitions were statically stretched at a speed equal to 5 mm/min and with a preset initial force of 20 kN. Measurement parameters such as the temperature of 23 °C and the relative humidity of 50% were taken into account. Tensile strength, Young’s modulus, elongation at break and impact strength were determined from the tests.

## 4. Conclusions

The use of dimeric imidazolium ionic liquids with aliphatic linkers of different lengths, constituting a new class of compounds known as gemini, proved to be an effective method for carrying out efficient functionalization of the cellulose nanofiller obtained by enzymatic hydrolysis with cellulase from the microscopic fungus *Trichoderma reesei*. The functionalization of the cellulose filler with innovative modifiers ensured the formation of a stable space structure between polysaccharide nanoparticles, limiting the creation of aggregates. It should also be noted that the synthesis of this type of gemini ionic liquids and their application to the functionalization of cellulose nanomaterials with their potential use as fillers in polymer matrices have not been described before.

Structural analyses (XRD) showed the presence of the first polymorphic variety in the tested materials—cellulose I, and the use of chemical modification using imidazolium ionic liquids did not result in changes in both the crystalline and amorphous structures of the polysaccharide. The effectiveness of the carried out functionalization of nanocellulose was confirmed by Fourier transform infrared (FTIR) spectroscopy, which proved changes in the chemical structure of the analyzed material by obtaining extra bands derived from carbonyl bonds (C=O), and in addition, the organic salts were bound to the hydroxyl groups (-OH) of cellulose and do not interfere with the glycosidic bonds of the polysaccharide. Dispersion research proved the spherical shape of cellulose particles for both the material before and after treatment with dimeric ionic liquids. In addition, nanocellulose after chemical modification was characterized by the presence of only a nanometric fraction. This is due to the acquisition of permanent covalent bonds between cellulose molecules because of the functionalization process.

Physicochemical studies of composite materials have shown interesting relationships between dispersion and morphological parameters of nanocelluloses obtained by cellulose functionalization and the formation of a supermolecular structure in the polypropylene matrix and the mechanical properties of nanocomposites. The study of phase transition proved that the chemical modification of the cellulose filler with imidazolium ionic liquids affects the nucleation activity of the produced nanocomposite systems. In turn, structural analysis of polymer composites showed that the type of cellulose nanofillers used had a significant effect on the generation of two polymorphic varieties of the polymer matrix. It was proven that the composite materials containing fillers of nanometric fraction were characterized by a significantly higher content of pseudohexagonal variety (β-PP), amounting to about 49%, than other analyzed systems. The obtained structural results correlate perfectly with the outcomes of mechanical properties by obtaining higher values of elongation at break, Young’s modulus and improved impact resistance of polysaccharide polymer composites after modification with dimeric imidazolium ionic liquids.

In conclusion, this article describes for the first time the synthesis as well as the functionalization of a nanocellulose polysaccharide with dimeric imidazolium ionic liquids with an alkyl linker of different lengths, constituting a new class of ionic liquids called gemini. In addition, this method has proven to be an effective technique for the efficient functionalization of cellulose through the formation of a stable space structure between polysaccharide molecules. It is noteworthy that this novel method of cellulose modification without the use of toxic organic solvents will directly affect the formation of the supermolecular structure of the polymer matrix and, consequently, the mechanical properties of the obtained polymer composites, which to increase their application potential.

## Figures and Tables

**Figure 1 ijms-23-15807-f001:**
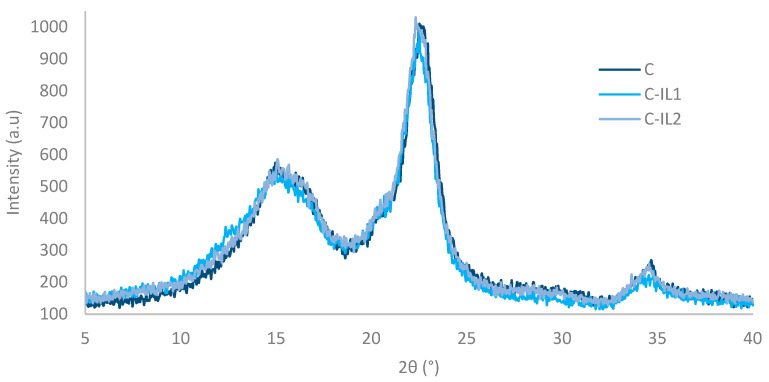
X-ray diffractograms for nanocellulose (C) and after functionalization with dimeric ionic liquids with shorter (C-IL1) and longer (C-IL2) alkyl linkers.

**Figure 2 ijms-23-15807-f002:**
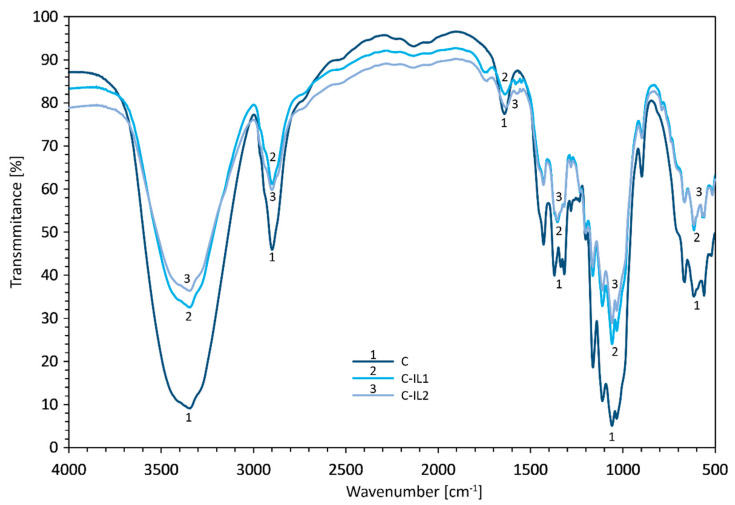
FTIR spectra for cellulose nanofiller before the chemical modification reaction (C) and after treatment with imidazolium ionic liquids with a shorter alkyl linker (C-IL1) and a longer linker (C-IL2).

**Figure 3 ijms-23-15807-f003:**
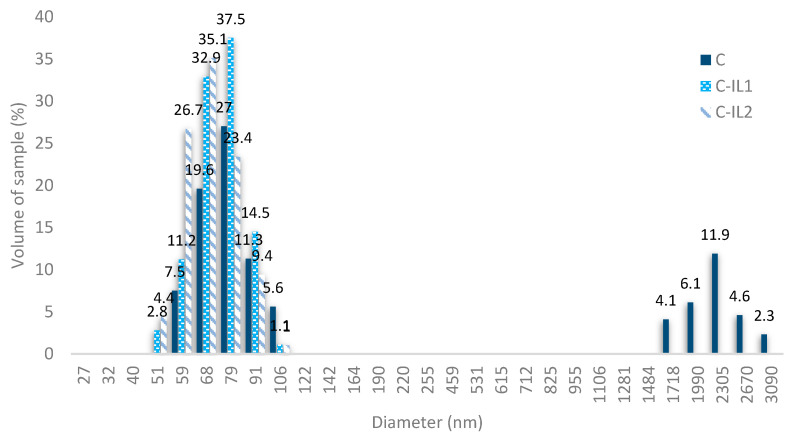
The particle size distribution of nanocellulose treated by enzymatic hydrolysis (C) and after functionalization with ionic liquids.

**Figure 4 ijms-23-15807-f004:**
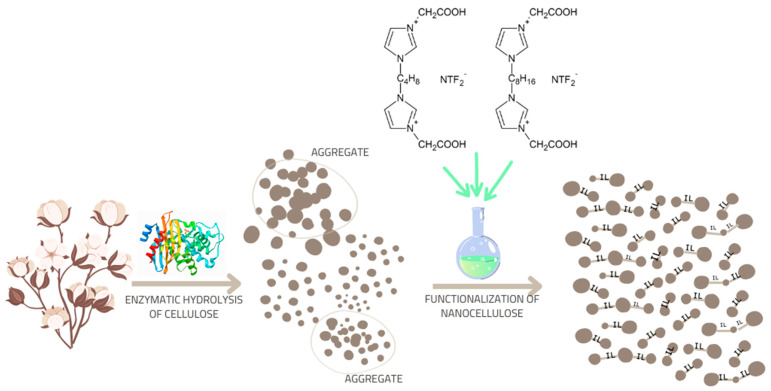
Mechanism of action of dimeric imidazolium ionic liquids against the applied cellulosic material.

**Figure 5 ijms-23-15807-f005:**
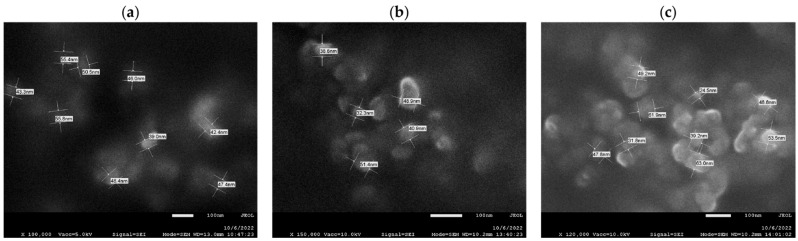
Scanning electron microscope images (C—(**a**)) nanocellulose obtained by enzymatic hydrolysis and nanocelluloses functionalized with ionic liquids with shorter (C-IL1—(**b**)) and longer (C-IL2—(**c**)) alkyl linkers.

**Figure 6 ijms-23-15807-f006:**
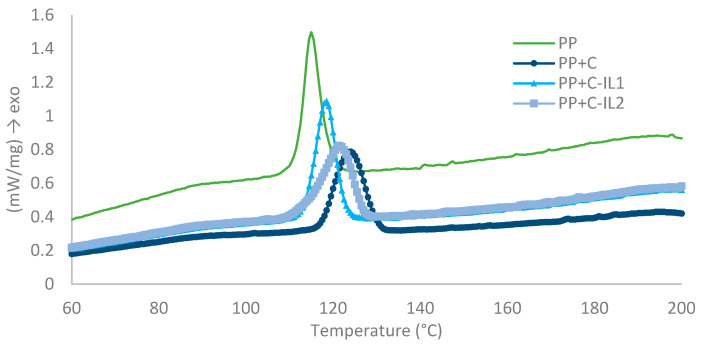
DSC thermograms of crystallization processes for polypropylene matrix and nanocomposites.

**Figure 7 ijms-23-15807-f007:**
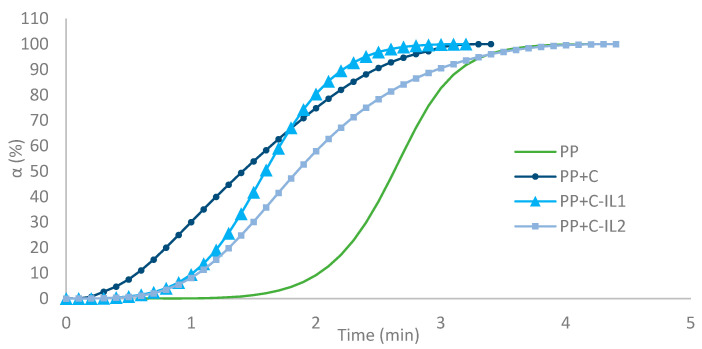
Phase conversion rate of unfilled polypropylene matrix and nanocomposite materials.

**Figure 8 ijms-23-15807-f008:**
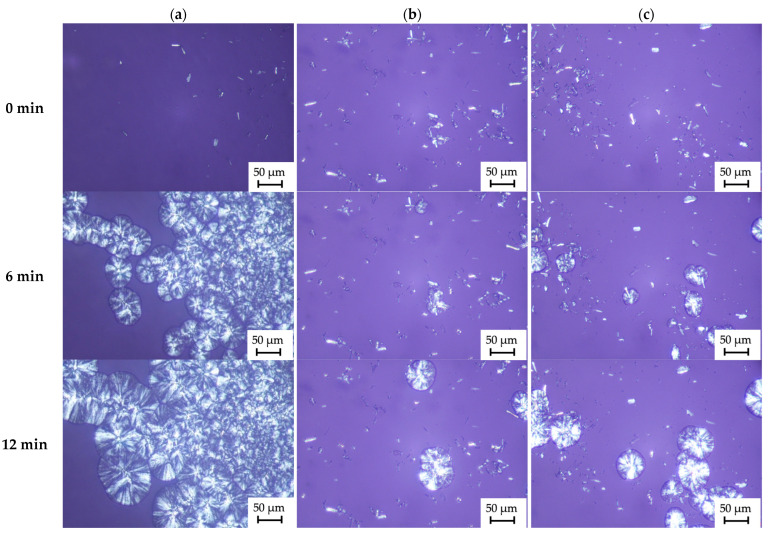
Optical microscope images of polymer nanocomposites (**a**) PP + C, (**b**) PP + C-IL1, (**c**) PP + C-IL2.

**Figure 9 ijms-23-15807-f009:**
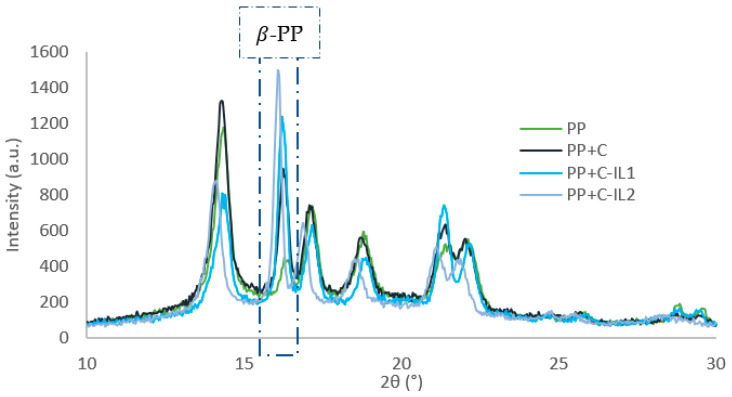
XRD patterns for polypropylene matrix and nanocomposite materials.

**Figure 10 ijms-23-15807-f010:**
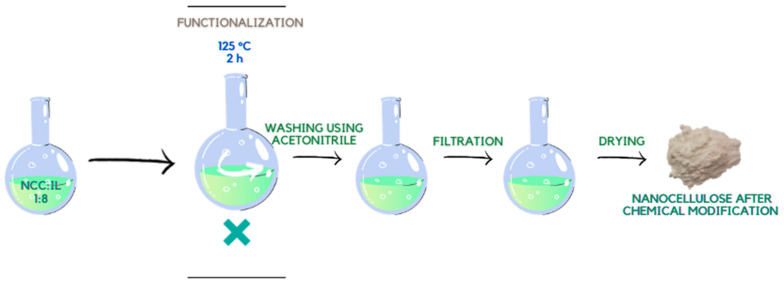
Scheme of chemical modification of nanocellulose using IL’s.

**Table 1 ijms-23-15807-t001:** The determined kinetic parameters (crystallization temperature (T_c_), melting temperature (T_m_) and half-time of crystallization (t_0.5_) and PLM analysis data (induction time and the growth rate of TCL) for the analyzed materials.

Material	Temperature of Crystallization (°C)	Temperature of Melting (°C)	Half-Time of Crystallization (min)	Induction Time (min)	The Growth Rate of TCL (μm/min)
PP	114.6	165.1	2.6	5	1.0
PP + C	123.8	165.2	1.4	1	2.7
PP + C-IL1	118.3	164.6	1.6	2.5	1.7
PP + C-IL2	121.3	165.1	1.8	4	1.3

**Table 2 ijms-23-15807-t002:** The content of the β-PP form and strength properties for polypropylene matrix and tested nanocomposites.

Material	Content of the β-PP Form (%)	Tensile Strength (MPa)	Young’s Modulus (GPa)	Elongation at Break (%)	Impact Strength (kJ/m^2^)
PP	9	29.0 (±0.12)	1.07 (±0.06)	303.3 (±27.3)	48.3 (±1.2)
PP + C	27	33.5 (±0.25)	1.39 (±0.11)	286.4 (±24.0)	29.3 (±2.5)
PP + C-IL1	44	37.3 (±0.31)	1.50 (±0.13)	377.6 (±19.4)	34.4 (±2.2)
PP + C-IL2	49	38.5 (±0.28)	1.66 (±0.10)	483.7 (±16.5)	37.8 (±1.8)

**Table 3 ijms-23-15807-t003:** Synthesized dimeric imidazolium ionic liquids.

Ionic Liquid	Chemical Name	Chemical Formula
**IL1** 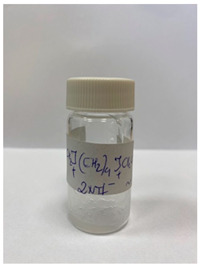	3,3′-[1,4-butane]-bis (1-carboxymethylimidazolium) di(bis(trifluoromethylsulfonyl)imide)	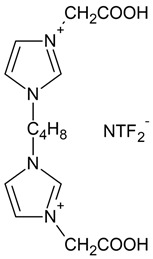
**IL2** 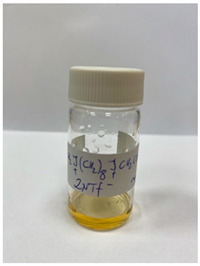	3,3′-[1,8-octane]-bis (1-carboxymethylimidazolium) di(bis(trifluoromethylsulfonyl)imide)	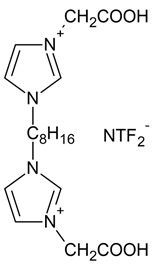

**Table 4 ijms-23-15807-t004:** A list of obtained nanofillers.

Sample	Material
C	Nanocellulose
C-IL1	Nanocellulose modified with IL1
C-IL2	Nanocellulose modified with IL2

## Data Availability

Not applicable.

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
