# Peer review of "Nanocellulose-Based Polymer Composites Functionalized with New Gemini Ionic Liquids"

_ijms, 2022, doi:10.3390/ijms232415807_

Round 1

Reviewer 1 Report

The manuscript provides an interesting new approach to the processing of composite nanomaterials.

A vast amount of data is presented. The data presentation is clear and logical and easy to follow. The background provided is more than sufficient to put the results in context, however, the authors could provide some more explanation of their results in this context:

Section 2.2.1 and Section 2.2.2:

L336: The authors state that: "Analyzing the presented results, it can be concluded that the chemical modification of the cellulose filler with dimeric imidazolium ionic liquids affects the nucleation activity of the produced nanocomposite systems." and L386: "The microscopic results are consistent with the previously discussed calorimetric tests, in which it was shown that nanocomposites with cellulose filler modified with imidazolium compounds with an ionic liquid with a shorter alkyl linker (PP+C-IL1) and with a longer linker (PP+C-IL2), showed lower nucleation activity than the system with enzymatically hydrolyzed nanofiller (PP+C),..."

Can the authors provide a more detailed explanation why the nucleation is affected in this way? It clearly demonstrated that it does occur, but the authors point towards conflicting answers in the literature (L360-L369), but fail to provide new insight based on their data. 

Section 2.2.4

The authors show a very interesting improvement in tensile strength, Young's modulus, and elongation in break and impact strength. It is far more common for nanofillers to increase Young's modulus and if the adhesion between polymer and filler is good, an increase in tensile strength. However, in most cases this leads to a reduction in elongation at break, meaning the filler seems to act as both a plasticizer as well as a reinforcing agent. Can the authors explain in more detail how this is possible? 

Minor considerations:

L761: please add a reference to a review article describing the dissolution of cellulose in ionic liquids. For example: 

Verma, Chandrabhan, et al. "Dissolution of cellulose in ionic liquids and their mixed cosolvents: A review." Sustainable Chemistry and Pharmacy 13 (2019): 100162.

Li, Yao, et al. "Towards a molecular understanding of cellulose dissolution in ionic liquids: Anion/cation effect, synergistic mechanism and physicochemical aspects." Chemical science 9.17 (2018): 4027-4043.

Pinkert, André, et al. "Ionic liquids and their interaction with cellulose." Chemical reviews 109.12 (2009): 6712-6728.

The quality of Figure 2 is poor. The number are too small to clearly read and the figure seems warped. 

L226: "Dispersion studies confirmed that performing chemical modification with ionic liquids has a significant effect on the dispersion properties of the resulting cellulose nanoparticles."

What exactly is the significance? is it statistical significance? Then please provide more details. 

The quality of the SEM images in Figure 5 is poor. The nanoparticles are blurry and hard to identify. Can the authors provide clearer images? 

Figure 7 is of too low resolution. Please update. 

Figure 8: The scale bars are very difficult to see and the number is not legible. 

Table 1 and Table 2 are formatted differently. Please be consistent with line thicknesses. 

Reviewer 2 Report

This manuscript by Daria Zielińska et al. reports on the use of novel ionic liquids for the modification of nanocellulose fillers. Such modification significantly affects the structures and properties of nanocellulose-polypropylene composites. Overall, this strategy may be useful for preparing green nanomaterials. However, there are several issues that the authors should address:

1. The figure quality needs to be improved. Some of figures have low resolution and the texts are difficult to read (e.g., Figure 4). Figure captions should also be more specific (e.g., Figure 1, 3).

2. Load-displacement curves from the mechanical testing should be provided in addition to Table 2.

3. The authors should also provide details of how impact strength/resistance is obtained. Based on the description, the ‘Impact Strength’ that the authors refer to are likely ‘Tensile toughness’. ‘Impact Strength’ should be based on ASTM standard impact tests and are different from ‘Tensile Toughness’.

4. The authors should carefully proofread the whole paper and correct the grammatical errors and typos (e.g., page 2, line 63).

Round 2

Reviewer 2 Report

The authors have addressed my comments. Some minor edits of formatting are needed (e.g. the captions of figure 2, 4 are far away from the figure and so are the Figure 5 labels).